# Integrative Bioinformatics and Experimental Validation Establish CCNB1 as a Potential Biomarker for Diagnosis and Prognosis in Colorectal Cancer

**DOI:** 10.3390/cimb47121026

**Published:** 2025-12-09

**Authors:** Yao Zou, Quan Zou, Zhen Li

**Affiliations:** 1Yangtze Delta Region Institute (Quzhou), University of Electronic Science and Technology of China, Quzhou 324000, China; zy1179520342@163.com (Y.Z.); zouquan@nclab.net (Q.Z.); 2School of Artificial Intelligence, Shenzhen University of Information Technology, Shenzhen 518172, China

**Keywords:** colorectal cancer, diagnosis, prognosis, survival, biomarker

## Abstract

Colorectal cancer (CRC) is a prevalent and lethal malignancy worldwide. Despite extensive research, core genes for diagnosis and prognosis in CRC remain to be fully elucidated. This study aims to identify novel gene biomarkers for CRC diagnosis and prognosis based on the GEO and TCGA datasets. Integration of TCGA and GEO datasets revealed 197 common differentially expressed genes (DEGs) between CRC tumor and normal samples. Functional enrichment analysis implicated these DEGs in biological processes and signaling pathways critical to CRC progression, including cell cycle regulation and nuclear division. Protein–protein interaction (PPI) network analysis identified 17 hub genes from DEGs, including *TROAP*, *CDKN3*, *CDCA3*, *UBE2C*, *CEP55*, *KIF11*, *CDC20*, *CCNA2*, *MCM4*, *CKS2*, *POLE2*, *MAD2L1*, *CCNB1*, *PTTG1*, *TPX2*, *TOP2A*, and *DLGAP5*. All 17 hub genes demonstrated high diagnostic value (AUC > 0.85), including *CCNB1* (AUC = 0.944). Based on the Cox proportional hazards regression, an 8-gene prognostic signature (*CLCA1*, *CCNB1*, *TPM2*, *MMP3*, *AOC3*, *CRYAB*, *CA4*, *GUCA2A*) effectively stratified patients by survival risk, with a 5-year AUC of 0.71. In vitro, *CCNB1* knockdown triggered cell cycle arrest, thereby suppressing the proliferation of colorectal cancer cells. This study validated *CCNB1* as a dual-purpose biomarker for CRC diagnosis and favorable prognosis, highlighting its potential utility in clinical management.

## 1. Introduction

Colorectal cancer (CRC) is a malignancy characterized by the abnormal proliferation of glandular epithelial cells in the rectum or colon [1]. Globally, over 1.9 million individuals are diagnosed with CRC each year, ranking as the third most prevalent cancer. In terms of mortality, it ranks as the second leading cause of death from cancer [2,3]. Proactive screening programs significantly reduce CRC incidence and mortality rates through diagnostic tools such as fecal occult blood testing (FOBT) and colonoscopy, which enable early detection of precancerous lesions. Standard therapeutic approaches for CRC include surgical resection, radiotherapy, chemotherapy, immunotherapy, and targeted small-molecule therapies [4,5]. Despite advances in the diagnosis and treatment, significant limitations in clinical management persist, and the underlying molecular mechanisms of CRC require further elucidation [6]. The molecular heterogeneity of CRC, primarily driven by genetic and epigenetic changes, poses a significant challenge to effective treatment. Therefore, identifying reliable biomarkers through transcriptomic analysis has become a critical research frontier.

The evolution of gene microarray and the next-generation sequencing (NGS) technologies enables the systematic identification of differentially expressed genes (DEGs) through integrative bioinformatics pipelines [7]. This approach combines transcriptome-wide profiling with statistical validation to find genes exhibiting significant dysregulation in disease states. Dysregulation of specific genes contributes critically to CRC pathogenesis by altering essential cellular processes, including proliferation and metastasis, thereby reducing patient survival [8]. Bioinformatic analyses have been widely employed to uncover prognostic biomarkers and elucidate molecular mechanisms in CRC. Previous transcriptomic studies have analyzed DEGs to uncover the molecular mechanisms driving CRC, identifying various gene signatures associated with disease progression and patient prognosis [8,9,10,11,12,13]. For instance, an investigation identified 10 hub genes with markedly elevated expression levels in CRC tissues, highlighting *CCNA2* as a potential oncogenic gene and novel biomarker in the disease process [8]. Another investigation proposed a set of 7 hub genes through bioinformatics approaches, suggesting their utility in exploring CRC development and underlying mechanisms [9]. Integrated bioinformatic and experimental investigations have revealed significant increase in the expression level of *CDC20* in CRC tumor compared to normal adjacent samples. Functional studies confirmed that depletion of *CDC20* led to a substantial reduction in the growth and expansion of CRC cell populations. These results suggest that *CDC20* acts as a critical oncogenic regulator in CRC [10]. Additionally, research has identified 10 prognostic genes, underscoring its potential utility as a novel tool for CRC patients [11]. Subsequent screening identified 6 hub genes potentially critical for CRC development. Survival analysis using the UALCAN database further revealed the significant correlation between high expression of *CCNA2* and *CCNB1* and poor prognosis [12]. Another study has also proposed a set of 4 hub genes as potential therapeutic targets for identifying and managing CRC patients [13]. However, the value of these DEGs for the early detection of CRC remains to be definitively established.

The development of bioinformatics databases and tools has greatly facilitated the exploration of cancer molecular mechanisms and identification of novel biomarkers [14]. Among these, the Gene Expression Omnibus (GEO), functions as primary global archive for public gene expression data [15]. This platform provides researchers with user-friendly tools for querying, downloading, and curating experimental datasets, particularly gene expression profiles across diverse biological conditions. The Cancer Genome Atlas (TCGA) is a major genomics program led by the National Human Genome Research Institute of the United States and the National Cancer Institute [16]. It comprehensively collects multi-omics data for various cancers, encompassing 33 cancer types and comprising over 20,000 samples. It also encompasses a wide range of data types, including genomic, transcriptomic, epigenetic, proteomic, and clinical data.

This study analyzed gene expression profiles from TCGA and GEO databases to identify DEGs by comparing CRC tumors with adjacent normal tissues. This study seeks to delineate core genes involved in CRC progression, evaluate their correlation with patient survival, and predict their role in tumorigenesis. These findings may serve as prognostic biomarkers for CRC risk stratification and reveal potential therapeutic targets for precision oncology interventions.

## 2. Materials and Methods

### 2.1. Data Collection

Two CRC gene expression arrays (GSE103512 from GPL13158, and GSE74602 from GPL6104) were systematically retrieved from the GEO database. The GSE103512 dataset comprised 7 CRC tissues and 7 adjacent normal samples, while the GSE74602 dataset contained 30 CRC tissues and 30 adjacent normal samples. In addition, COAD (Colon Adenocarcinoma) gene expression datasets and the matched survival metadata were from the TCGA database. A total of 514 tissue samples, including 473 from CRC tissues and 41 from adjacent normal tissue, were included in the analyzed cohort of the TCGA-COAD study. The scRNA-seq data of one CRC tumor sample was from GSE277669.

### 2.2. Identification of DEGs Shared by the Three Datasets

To detect DEGs, we applied the limma package (v3.62.2) in R v4.4.1 to analyze the microarray and TCGA datasets [17]. After quantile normalization, we applied empirical Bayes moderated *t*-tests to identify DEGs between tumor and normal samples within each dataset. Volcano plots were generated via the ggplot2 package (v3.5.1) to visualize differential genes within each dataset.

Venn diagrams (https://www.bioinformatics.com.cn/static/others/jvenn/, accessed on 29 November 2025) were used to visualize overlapping differentially expressed genes among the three datasets. A combined threshold of *p*-value below 0.05 and absolute log2 fold change greater than 1 was applied to identify consensus DEGs present in all three datasets.

### 2.3. Functional Enrichment for the DEGs Based on the GO and KEGG Databases

We performed functional enrichment analysis of DEGs on the GO and KEGG databases with the clusterProfiler R package (v4.14.6) [18] and visualized the results using ggplot2 (v3.5.1) in R v4.4.1.

### 2.4. Identification and ROC Curve Analysis of Hub Genes

The network topology of protein–protein interactions (PPI) for the DEGs was generated from the STRING database [19], and rendered for analysis within the Cytoscape v3.10.3 [20], where nodes correspond to proteins and edges denote interactions. The MCODE module within the Cytoscape environment was utilized to analyze network density and screen for central node genes in the most significant modules. The cytoHubba plugin (version 0.1) was employed to detect hub genes from the constructed PPI network [21]. We evaluated the diagnostic efficacy of these genes by conducting ROC curve analysis, determining their ability to discriminate between CRC tumor and normal samples.

### 2.5. Survival Analysis

We conducted Kaplan–Meier method and univariate Cox regression analysis to identity survival-related candidate genes from DEGs [22]. The intersection of genes screened by those two methods with *p* < 0.05 was considered as survival-related candidate genes. Leveraging the expression of candidate genes and overall survival data, we computed prognostic risk scores using a multivariate Cox proportional hazards model. It was defined as: Risk score = Σ (coefficient_i × expression_i) (where expression_i represents the gene expression level and coefficient_i denotes the regression coefficient). Kaplan–Meier survival analysis was employed to evaluate the overall survival (OS) disparity between high- and low-risk CRC patient cohorts dichotomized by the median risk score. The predictive accuracy was quantified by generating time-dependent ROC (timeROC) curves, facilitated by the timeROC package (version 0.4) [23].

### 2.6. Survival-Related Hub Genes Expression in scRNA-Seq Data of CRC Tumors

Single-cell transcriptomics (scRNA-seq) data from a CRC tumor sample was processed using Seurat (v5.2.1) [24]. Initial quality control excluded cells with mitochondrial gene percentages exceeding 10% (mt_percent < 10). Subsequent analysis included normalization, detection of highly variable genes, and principal component analysis (PCA). For cell clustering, we applied the FindNeighbors and FindClusters functions, followed by dimensionality reduction and visualization via t-SNE and UMAP [25]. The FindAllMarkers function was applied to compute cluster-specific marker genes using predefined thresholds: a minimum 25% cellular detection rate within a cluster, coupled with *p* < 0.05 significance. Cell types were annotated by matching expression profiles to established markers in the CellMarker (http://117.50.127.228/CellMarker, accessed on 29 November 2025) [26]. Finally, UMAP plots and violin plots were applied to visualize the expression of survival-related hub genes in each cell type.

### 2.7. Cell Culture and siRNA Transfection

The human colorectal adenocarcinoma cell line SW480 was purchased from iCell Bioscience Inc. (Shanghai, China). The human CRC cell line SW480 was cultured in medium with 10% FBS. For experiments, cells were harvested and transferred to 6-well plates during their logarithmic growth phase.

We performed siRNA transfection to silence *CCNB1*, introducing CCNB1-targeting siRNA into cells at 80% confluency. The universal negative control group consisted of a siRNA with a non-specific sequence (si-NC), and an empty plasmid served as the blank control (control). For each well, transfection was performed using si-CCNB1 or si-NC (CCNB1-s: CCUGGCUAAGAAUGUAGUC; CCNB1-a: GACUACAUUCUUAGCCAGG) at a concentration of 10 mg/mL with Hieff Trans Liposomal 2000 (Yeasen, Shanghai, China). Cells were harvested 24–48 h post-transfection for detection of transfection efficiency.

### 2.8. qPCR

Total RNA extraction from the SW480 cells was performed with Trizol reagent (R0016, Beyotime Biotechnology, Shanghai, China) in accordance with the manufacturer’s instructions. To generate cDNA libraries, reverse transcription was conducted utilizing the RevertAid RT kit (Thermo Fisher Scientific, Shanghai, China). On the Applied Biosystems platform, qPCR amplifications were carried out employing the 2 × SYBR Green qPCR Master Mix (B21203, Selleck, Shanghai, China). The expression levels of *CCNB1* were quantified relative to the endogenous control GAPDH using the 2^−ΔΔCt^ method.

### 2.9. Western Blot

After collecting the SW480 cells, the total protein was extracted using RIPA lysis buffer (P0013B, Beyotime). Quantification of total protein was performed employing the BCA method using a commercial kit (P0010, Beyotime). Following electrophoretic separation, the PVDF membrane with proteins was subsequently incubated for 1 h with blocking solution, and then underwent incubation with the primary antibody overnight at 4 °C. The secondary antibody incubation was carried out at room temperature for 1 h, preceded by and followed with washes in TBST buffer. Following TBST washes, the membrane was treated with ECL luminescent substrate (PK10003, Proteintech, Wuhan, China), and signals were detected using a gel imaging system (Jingyi Technology, Guangzhou, China). GAPDH (Abcam ab8245, Shanghai, China) was used as an internal control.

### 2.10. Cell Cycle Assay

The cell cycle was analyzed using flow cytometry with an Annexin V Alexa Fluor 488/PI Apoptosis Detection Kit (CA1020, Solarbio, Beijing, China). Cells were digested with EDTA-free trypsin and centrifuged to obtain a pellet. The binding buffer was diluted 1:9 with deionized water and used to resuspend the cell pellet. For staining, cells were resuspended in a 100 µL aliquot and labeled with 5 µL of Alexa Fluor 488-conjugated Annexin V in a flow cytometry tube. Cells were then incubated in the dark for 5 min at room temperature. Subsequently, 5 µL from a 20 µg/mL propidium iodide solution was supplemented, followed by the addition of 400 µL PBS. Flow cytometric analysis was conducted without delay on a CytoFLEX system (Beckman Coulter Life Sciences, Miami, FL, USA).

### 2.11. Cell Viability Assay (CCK-8)

A cell suspension was adjusted to 3000 cells per 100 μL and seeded into a 96-well plate, followed by overnight incubation. The culture medium was then aspirated and discarded. Following the addition of 100 μL CCK-8 detection reagent (CK04, Dojindo, Shanghai, China) to each well, a 2-h incubation at 37 °C was carried out to achieve optimal colorimetric signal development. The optical density (OD) value at 450 nm (OD450) was measured at the time points of 0, 24, 48, and 72 h.

### 2.12. Statistical Analysis

The study utilized the R platform (v4.4.1) for all statistical evaluations. The Wilcoxon rank-sum test was employed to compare independent groups [27], with a *p*-value below the 0.05 threshold denoting statistical significance.

## 3. Results

### 3.1. DEGs Analysis

The dataset GSE103512, GSE74602, and TCGA-COAD identified 416, 1486, and 7828 DEGs, respectively (Appendix A). A Venn diagram identified 197 overlapping DEGs across all three datasets, consisting of 68 upregulated and 129 downregulated genes (Figure 1a, Appendix A).

### 3.2. Functional Enrichment for the DEGs

Based on GO enrichment analysis of DEGs, the significant enrichment in biological processes (BP) for upregulated DEGs were primarily linked to nuclear division, nuclear chromosome segregation, sister chromatid segregation, and microtubule cytoskeleton organization involved in mitosis. At the cellular component (CC) level, upregulated DEGs predominantly localized to the spindle, spindle pole, and specific kinase complexes encompassing cyclin-dependent and serine/threonine protein kinase holoenzymes. In addition, molecular function (MF) analysis indicated the most significant enrichment for regulators of cyclin-dependent protein serine/threonine kinase activity, highlighting a core function in cell cycle control (Appendix A). Enrichment profiling of downregulated DEGs highlighted muscle system processes and contraction (BP), encompassed the collagen-rich extracellular matrix and contractile structures (CC), and centered on actin binding functionality (MF) (Appendix A).

KEGG analysis further revealed that upregulated DEGs were significantly enriched in the cell cycle, IL-17 signaling pathway, DNA replication, and rheumatoid arthritis (Figure 1b, Appendix A).

### 3.3. Identification of Hub Genes in PPI Network and ROC Curve Analysis

A PPI network was built using STRING, comprising 159 gene nodes and 1036 edges. DEGs were visually represented, with 53 upregulated and 106 downregulated genes being color-coded in red and blue, respectively (Figure 2a). Subsequent analysis via Cytoscape cytoHubba identified 17 hub genes among the 159 nodes: *TROAP*, *CDKN3*, *CDCA3*, *UBE2T*, *CEP55*, *KIF11*, *CDC20*, *CCNA2*, *MCM4*, *CKS2*, *POLE2*, *MAD2L1*, *CCNB1*, *PTTG1*, *TPX2*, *TOP2A*, and *DLGAP5* (Figure 2b). Notably, all hub genes were upregulated and highlighted in yellow (Appendix A).

To evaluate the diagnostic potential of the 17 hub genes in CRC, we performed ROC curve analysis based on the TCGA cohort. As shown in Appendix A, all hub genes exhibited robust diagnostic performance, with AUC values as follows: *TROAP* (0.974), *CDKN3* (0.906), *CDCA3* (0.954), *UBE2T* (0.950), *CEP55* (0.949), *KIF11* (0.874), *CDC20* (0.890), *CCNA2* (0.908), *MCM4* (0.938), *CKS2* (0.942), *POLE2* (0.883), *MAD2L1* (0.934), *CCNB1* (0.944), *PTTG1* (0.930), *TPX2* (0.973), *TOP2A* (0.922), and *DLGAP5* (0.898).

### 3.4. Prognostic Model Construction for Survival Prediction

A total of 16, and 21 survival related genes were identified by the Kaplan–Meier survival analysis (Appendix A, Logrank test, *p*-value < 0.05), and the univariate Cox analysis (Appendix A, *p*-value < 0.05), respectively. The intersection of genes selected by both methods yielded 8 survival-associated genes: *CLCA1*, *CCNB1*, *TPM2*, *MMP3*, *AOC3*, *CRYAB*, *CA4*, and *GUCA2A*. The Kaplan–Meier curves for those 8 genes were shown in Appendix A. Using these 8 genes and overall survival data, we developed a multivariate Cox proportional hazards model to calculate prognostic risk scores. Risk quantification was performed using the equation: Risk Score = (−0.0537 × Expression *CLCA1*) + (−0.2222 × Expression *CCNB1*) + (0.0936 × Expression *TPM2*) + (0.1112 × Expression *MMP3*) + (0.0025 × Expression *AOC3*) + (0.0591 × Expression *CRYAB*) + (−0.0011 Expression *CA4*) + (−0.0411 × Expression *GUCA2A*).

As shown in Figure 3a,b, CRC patients with higher risk scores exhibited increased mortality rates. Figure 3c displays the expression heatmap of 8 prognostic genes across groups stratified. Kaplan–Meier analysis demonstrated a pronounced disparity in survival, with the high-risk cohort exhibiting a significantly diminished probability of survival compared to low-risk cohort (Figure 4a). Time-dependent ROC curves validated the prognostic signature, with AUC values of 0.60 (1-year), 0.61 (3-year), and 0.71 (5-year) in the training set (Figure 4b), and 0.71 (1-year), 0.65 (3-year), and 0.72 (5-year) in the test set (Figure 4c).

### 3.5. Expression of the Prognostic Hub Gene CCNB1 in CRC scRNA-Seq Dataset

Intersection analysis of eight prognostic genes and seventeen hub genes identified *CCNB1* as a prognostic hub gene. To explore the cellular role of *CCNB1* in CRC, we analyzed scRNA-seq data from a tumor sample. After quality control, 1676 single cells were annotated using lineage markers and subsequently clustered into 12 clusters (Figure 5a). Cluster-specific marker genes were calculated using the FindAllMarkers algorithm, and the top ten marker genes per cluster were visualized in a heatmap (Appendix A). UMAP and violin plots demonstrated that *CCNB1* was specifically and highly expressed in epithelial cell cluster 6 (Figure 5b and Appendix A). Furthermore, this cluster (epithelial cluster 6) exhibited the highest cell cycle activity, as indicated by elevated S/G2M scores, which aligns with its high *CCNB1* expression (Appendix A).

### 3.6. The siRNA Transfection Effects on CCNB1 Expression, Cell Cycle, and Proliferation in SW480 Cells

qPCR analysis was performed to assess the transcriptional level of *CCNB1* in SW480 cells. Transfection with si-CCNB1 significantly reduced *CCNB1* mRNA expression by approximately 50% compared to control and si-NC groups (Figure 6a, *p* < 0.01). Furthermore, a decrease in CCNB1 protein levels was observed via Western blot in the si-CCNB1 group relative to the two control conditions (Figure 6b). The flow cytometry quadrant analysis revealed that si-CCNB1 treatment increased the proportions of early apoptotic (LR) and late apoptotic (UR) cells, while correspondingly reducing the viable cell (LL) population (Appendix A). This indicated that si-CCNB1 triggered both cell cycle disruption and apoptotic death. Assessment of SW480 cell proliferation via the CCK-8 assay revealed that silencing *CCNB1* effectively suppressed proliferation compared to both controls (Figure 6c,d, *p* < 0.01).

## 4. Discussion

### 4.1. Identification of 17 Hub Genes with High Diagnostic Potential in CRC Pathogenesis

Our study identified 197 DEGs (68 upregulated, 129 downregulated) between CRC and normal tissues by integrating two GEO microarray datasets and TCGA RNA-seq data. Functional enrichment analysis revealed that the upregulated DEGs were significantly enriched in nuclear division, nuclear chromosomal segregation, sister chromatid segregation, and microtubule cytoskeleton organization involving spindle and spindle pole formation. Those processes are implicated in chromosomal instability (CIN), a key driver of CRC tumorigenesis and progression [28,29,30,31]. KEGG analysis revealed upregulated DEGs enrichment in cell cycle regulation, IL-17 signaling, and DNA replication pathways, aligning with previous studies [32,33,34]. These findings collectively demonstrated that the upregulated DEGs drive CRC oncogenesis through the activation of multiple proliferative signaling pathways.

By constructing a PPI, we identified 17 hub genes, all of which were upregulated. These 17 hub genes can be categorized into 4 functional classes, consistent with the functional enrichment results (GO and KEGG) of the upregulated DEGs:(1).Cell cycle regulation genes: *CCNA2*, *CCNB1*, *CDKN3*, *CDC20*.

Multiple investigations have documented elevated expression of *CCNA2* and *CCNB1* in colorectal cancer [9,12]. Cyclin A2 (*CCNA2*) regulates cell cycle progression by binding cyclin-dependent kinases (CDKs) to drive G1/S and G2/M phase transitions. Meanwhile, *CCNB1* encodes mitotic regulatory proteins essential for controlling G2/M transition. Inhibition of *CCNB1* suppresses tumor growth and triggers cell cycle disruption as well as apoptotic death in CRC and gastric cancer [9]. Additionally, *CDKN3* and *CDC20* exhibit high expression in CRC tissues, correlating with disease diagnosis and prognostic evaluation [10,35].

(2).Mitosis-associated genes: *KIF11*, *TPX2*, *DLGAP5*, *TROAP*, *CEP55*.

*KIF11*, a critical mitotic motor protein, facilitates spindle formation and orientation during early mitosis. In colorectal cancer, elevated *KIF1*1 expression correlates with well-differentiated histology and improved patient prognosis [36]. Similarly, *TPX2* demonstrates elevated expression level in CRC tissues [37]. Discs Large Homolog Associated Protein 5 (*DLGAP5*), contributes to mitotic spindle assembly and stabilization. Its dysregulation influences proliferative and migratory capacities across multiple malignancies [38,39]. *TROAP* exhibits oncogenic properties in CRC and ovarian adenocarcinoma, where its overexpression drives malignant progression and tumor aggressiveness [40]. Centrosomal Protein 55 (*CEP55*) modulates mitotic fidelity and PI3K/AKT signaling, with dysregulation linked to various cancers and inflammatory disorders [41].

(3).DNA replication and repair genes: *MCM4*, *POLE2*, *TOP2A*.

Emerging evidence indicates that *MCM4* is overexpressed in multiple malignancies, including hepatocellular carcinoma, and esophageal tumor [42,43,44]. Such dysregulation may drive cell cycle aberrations, thereby accelerating uncontrolled proliferation and facilitating tumorigenesis and metastasis. Separately, the DNA polymerase epsilon 2 accessory subunit (*POLE2*), which modulates cellular proliferation mechanisms, has been identified as an overexpressed gene in colorectal cancer [45]. Concurrently, *TOP2A* upregulation during tumor development correlates with therapeutic response to pharmacologic interventions in CRC [46].

(4).Chromosomal stability maintenance genes: *MAD2L1*, *PTTG1*, *CDCA3*, *UBE2T*, *CKS2*.

*MAD2L1*, *PTTG1*, and *CKS2* were identified as core genes implicated in CRC pathogenesis [11]. Among these, *MAD2L1* and *PTTG1* may function as stage-dependent biomarkers for CRC progression [47]. *CKS2* overexpression correlates with aggressive tumor phenotypes in CRC, suggesting its role in driving malignancy [48]. Separately, cell division cycle-associated protein 3 (*CDCA3*), a critical regulator of mitotic entry, exhibits dysregulated upregulation in CRC tissues [49]. Ubiquitin-conjugating enzyme E2T (*UBE2T*), a key mediator of DNA damage repair and genomic stability [50], is overexpressed in CRC patients and promotes tumor progression through ubiquitin-mediated degradation of p53 [51].

Given the established role of 17 hub genes in various cancers, we evaluated their diagnostic potential. The ROC curve analysis demonstrated high diagnostic accuracy across all 17 genes, supporting their strong clinical utility for CRC detection.

### 4.2. An 8-Gene Prognostic Risk Model for Survival Prediction

We developed and validated an 8-gene prognostic risk model with significant predictive accuracy for CRC patient survival. This model effectively stratified patients into distinct high- and low-risk cohorts with significant divergence in clinical outcomes. Time-dependent ROC curves demonstrated its predictive capability, with AUCs of 0.60 (1-year), 0.61 (3-year), and 0.71 (5-year) in the training set, and 0.71 (1-year), 0.65 (3-year), and 0.72 (5-year) in the test set.

By employing bioinformatics approaches, our study facilitated the discovery of prognostic signatures relevant to CRC management. Similar studies have developed prognostic signatures leveraging CRC-related DEGs to predict patient outcomes. For instance, researchers integrated bioinformatics analyses to identify 12 pyroptosis-associated genes, ultimately deriving a 4-gene signature that accurately predicted survival outcomes in CRC cohorts [52]. Similarly, integrated single-cell and bulk transcriptomics were used to delineate anoikis-related molecular profiles, revealing a 10-gene prognostic signature strongly correlated with clinical prognosis; this model achieved an AUC of 0.755 in time-ROC analysis for 5-year survival prediction [53]. In parallel, ferroptosis-related DEGs were investigated using WGCNA combined with univariate and LASSO Cox regression, leading to the establishment of an 11-gene signature [54]. A prognostic risk model was built using 11 autophagy-related hub genes, suggesting their therapeutic potential for CRC. The predictive capability of our model was evaluated via time-ROC analysis, with AUC scores above 0.6 for 1-, 3-, and 5-year survival predictions [55]. A separate investigation yielded a 9-gene signature with robust predictive performance for CRC survival (5-year AUC: 0.741), a finding that was confirmed in independent cohort analyses [56].

### 4.3. Functional Validation of the Prognostic Hub Gene CCNB1

Our findings demonstrated that *CCNB1* is markedly overexpressed in CRC tissues and that its expression level is significantly associated with improved prognosis. Most importantly, in vitro functional experiments provided direct evidence that inhibition of *CCNB1* expression attenuated cellular proliferation in CRC cells. These findings underscore the pivotal role of *CCNB1* in driving CRC pathogenesis, highlighting its promising potential in CRC patients.

Cell cycle-associated protein B1 (*CCNB1*) is a crucial cyclin B family member that functions as a principal initiator and imposes stringent quality control during mitosis. *CCNB1* exhibits peak expression during the G2/M phase transition to ensure proper cell division. This conserved cyclin regulates the cell cycle by forming functional complexes with specific cyclin-dependent kinases (CDKs) [57]. The association between *CCNB1* and CDK1 facilitates the catalytic activation of CDK1 via phosphorylation at its Thr161 residue, culminating in the formation of the active heterodimeric complex recognized as Maturation-Promoting Factor (MPF). MPF drives G2-to-M phase progression and mitosis [57,58]. The degradation of *CCNB1*, which inactivates the CDK1, is prerequisite for mitotic exit and entry into the subsequent cell cycle. The precise timing of mitotic events hinges on the tightly controlled regulation of the *CCNB1*-CDK1 axis, as it determines the transition into mitosis, enforces G2 phase arrest, or permits cycle bypass in response to specific signals.

Previous studies have reported elevated *CCNB1* expression across multiple cancers [59,60], a finding that is in line with our observations in CRC. Substantial evidence confirms elevated *CCNB1* expression across tumor types enhances malignant proliferation and metastasis [61,62]. For instance, forkhead box protein M1 (FOXM1) activates *CCNB1*, triggering accelerated hepatocellular carcinoma (HCC) cell proliferation [61]. Conversely, in pancreatic cancer, downregulation of *CCNB1* triggers p53 signaling, resulting in suppressed tumor growth and concomitant induction of cellular senescence [62]. Additionally, Chk1-induced *CCNB1* overexpression promotes colorectal tumor growth [63]. Notably, disruption of the *CCNB1*/*CDK1* complex suppresses carcinogenesis and metastatic invasion, as demonstrated in recent studies [64].

The association between high *CCNB1* expression and improved prognosis in CRC patients presents an intriguing finding. In this study, prognostic analysis based on *CCNB1* levels revealed a significantly higher survival probability for the high-expression group. In addition, the negative regression coefficient for *CCNB1* in our prognostic model indicated an inverse correlation between its expression and risk scores, where elevated *CCNB1* predicted diminished mortality risk. These results aligned with some previous reports [65,66]. However, univariate and multivariate analyses revealed that *CCNB1* does not serve as an independent prognostic gene for CRC patients. *CCNB1* dysregulation represents an early tumorigenic event, with its expression patterns intricately linked to cancer progression. Previous studies have investigated the relationships between the *CCNB1* expression and prognosis in various cancers [60,67,68,69]. *CCNB1* overexpression correlates with adverse prognosis in diverse tumors, notably hepatocellular carcinoma, breast cancer, and bladder cancer [60,66,69]. Conversely, studies in lymphoma, thymoma and CRC have reported favorable outcomes associated with elevated *CCNB1* expression [65,66,69,70], highlighting its tumor-type-specific prognostic role. While an association between *CCNB1* expression and improved prognosis in CRC has been established, the underlying molecular mechanisms remain incompletely understood, thus representing a key focus for future studies.

### 4.4. Research Innovations, Limitations, and Future Research Directions

In comparison with previous studies, a key advancement of our work is the comprehensive assessment of *CCNB1*’s dual clinical utility. Our analysis demonstrated the exceptional diagnostic power of *CCNB1*, achieving an AUC of 0.944. More importantly, we incorporated *CCNB1* into a multi-gene prognostic signature and validated its independent value for patient risk stratification using survival analysis, which provides a more nuanced tool beyond mere expression correlation. In addition, a significant contribution that distinguishes our work is the experimental validation of *CCNB1*’s biological function in CRC progression. Through in vitro experiments, we provided direct evidence that *CCNB1* promotes CRC cell proliferation, thereby bridging the gap between bioinformatic prediction and biological causality.

In summary, our study identified upregulated *CCNB1* expression in CRC, with ROC analysis validating its high diagnostic accuracy. Additionally, survival analysis and prognostic modeling demonstrated that elevated *CCNB1* levels correlated with increased survival probability and further indicated diminished mortality risk. In conclusion, our findings establish *CCNB1* as a robust diagnostic biomarker and a favorable prognostic indicator in CRC. Despite limitations including the modest predictive accuracy, the confinement of *CCNB1* validation to proliferation/apoptosis, and restricted generalizability from a single scRNA-seq sample, this study provides valuable insights into CRC clinical management. In future research, we will prioritize collecting all available clinicopathological data (such as TNM stage and CEA levels), which will be integrated with genetic signatures to construct a more robust predictive model with enhanced performance. Additional validation including downstream pathway activity (e.g., CDK1 activation or key cell cycle checkpoint markers) should be incorporated to provide a more comprehensive understanding of *CCNB1*’s role. Moreover, larger cohorts are warranted to validate and extend our observations.

## Figures and Tables

**Figure 1 cimb-47-01026-f001:**
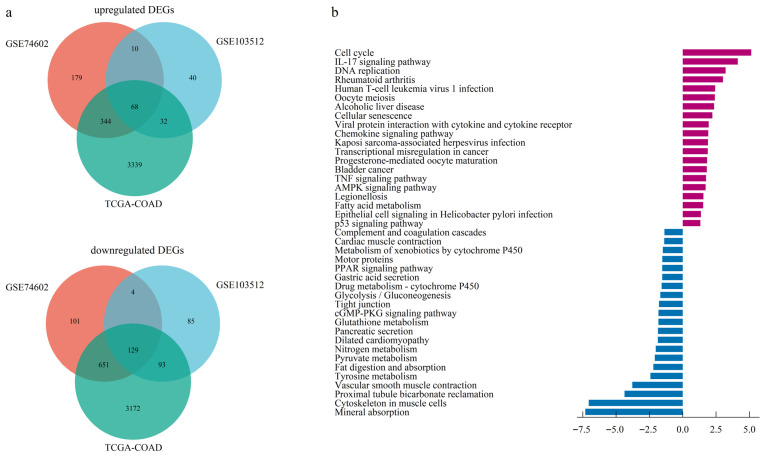
Identification and functional enrichment of differentially expressed genes (DEGs). (**a**) Venn diagram illustrating the intersection of up-regulated and down-regulated DEGs identified from two GEO datasets and one TCGA dataset. (**b**) Barplot of KEGG pathway enrichment analysis for the consensus DEGs.

**Figure 2 cimb-47-01026-f002:**
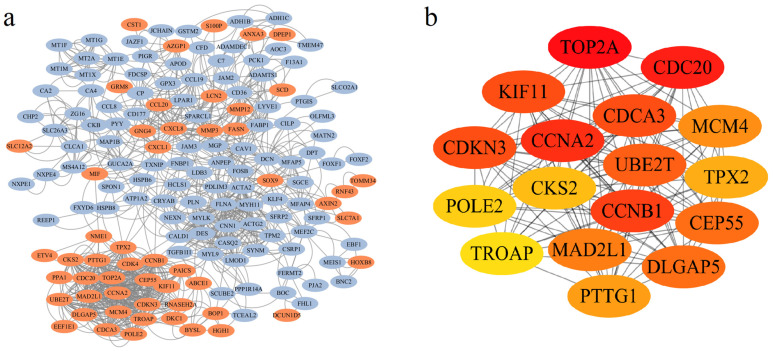
Analysis of the Protein–Protein Interaction (PPI) network topology and identification of hub genes. (**a**) Visualization of the PPI network generated from STRING database and rendered in Cytoscape, with nodes colored based on their differential expression status (red: up-regulated; blue: down-regulated). (**b**) Identification of 17 hub genes based on the MCC score using the CytoHubba plugin. The color gradient from yellow to red represents increasing node centrality values, with darker red nodes indicating higher importance within the network.

**Figure 3 cimb-47-01026-f003:**
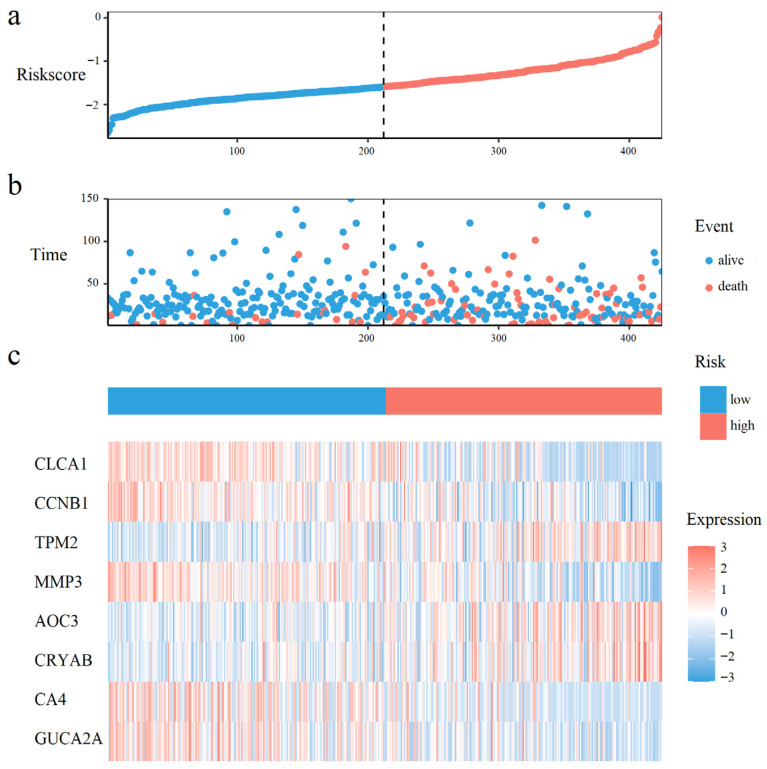
Prognostic risk distribution, patient survival status, and gene expression patterns. (**a**) The risk curve, ordered by ascending risk score. (**b**) Survival status scatter plot, where each point represents a patient. (**c**) A heatmap of prognostic gene expression in the risk-ordered cohort.

**Figure 4 cimb-47-01026-f004:**
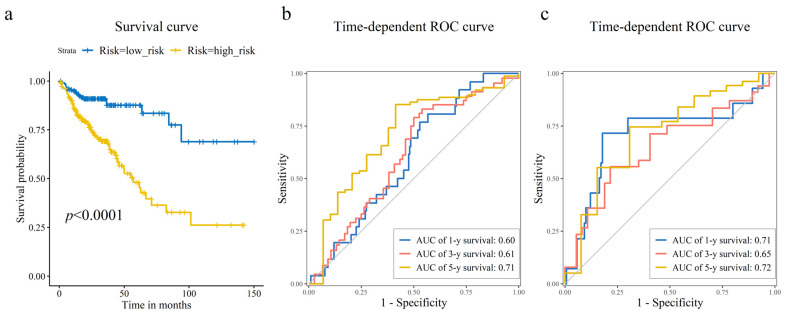
Prognostic risk model assessment and validation. (**a**) Kaplan–Meier survival curves comparing the high-risk and low-risk groups stratified by the median risk score. Log-rank test was used to assess the statistical significance of the difference in survival distributions. Time-dependent receiver operating characteristic (ROC) curves illustrating the model’s accuracy for predicting 1-year, 3-year, and 5-year overall survival in the (**b**) training, and (**c**) test sets. The corresponding area under the curve (AUC) values are displayed, showing the model’s sustained accuracy over time.

**Figure 5 cimb-47-01026-f005:**
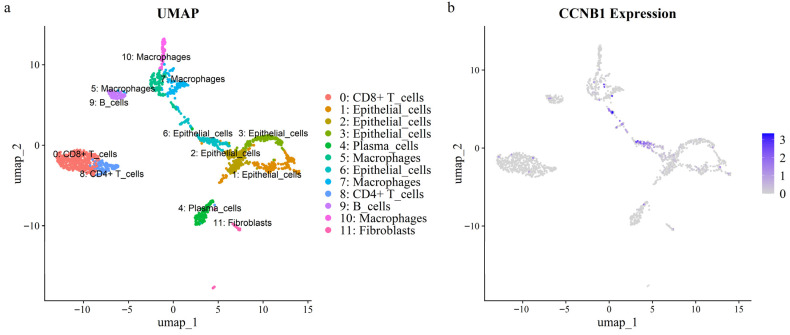
UMAP visualization of a CRC single-cell RNA dataset. (**a**) Cellular clusters identified within the dataset. (**b**) Expression levels of the *CCNB1* gene across the identified clusters.

**Figure 6 cimb-47-01026-f006:**
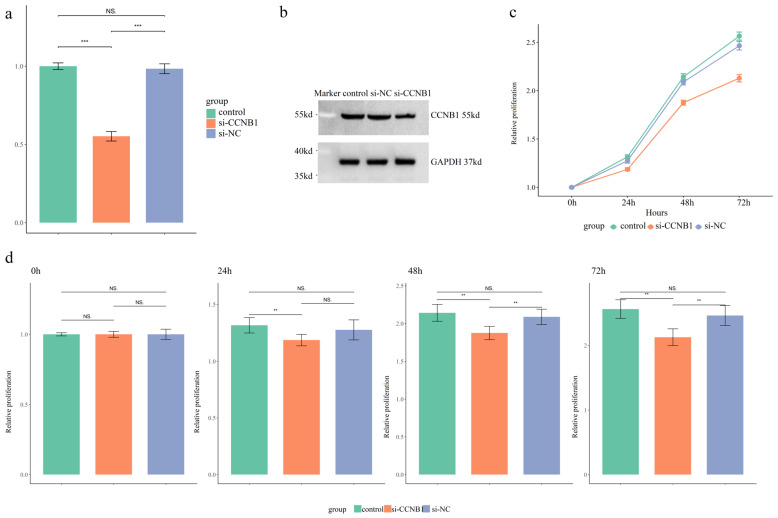
Functional Validation of *CCNB1* in SW480 Cells. (**a**) Quantitative real-time PCR (qRT-PCR) analysis of *CCNB1* mRNA expression levels across three experimental groups. (**b**) Western blot analysis of CCNB1 protein expression levels in the three groups. (**c**) Assessment of cell proliferative capacity over time via CCK-8 assay, presented as a growth curve. (**d**) Column graph comparing the proliferative capacity of SW480 cells across the three groups at specific time points (0, 24, 48, and 72 h). The results are expressed as mean ± SD. ** *p *< 0.01, *** *p *< 0.001, NS *p *> 0.05.

## Data Availability

The data presented in this study are openly available in the NCBI Gene Expression Omnibus (GEO) database at https://www.ncbi.nlm.nih.gov/, under the Series GSE103512, GSE74602, GSE277669. Further inquiries can be directed to the corresponding author.

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
