# Peer review of "Integrative Bioinformatics and Experimental Validation Establish CCNB1 as a Potential Biomarker for Diagnosis and Prognosis in Colorectal Cancer"

_cimb, 2025, doi:10.3390/cimb47121026_

Round 1
Reviewer 1 Report
Comments and Suggestions for Authors
This manuscript explores the identification of CCNB1 as a diagnostic and prognostic biomarker for colorectal cancer (CRC) through integrative bioinformatics analysis and experimental validation. The study combines GEO and TCGA datasets, constructs a prognostic model, and validates CCNB1’s functional role using in vitro assays. The topic is relevant to molecular oncology and aligns with the journal’s scope, given its emphasis on molecular mechanisms and translational potential. However, several critical issues regarding novelty, methodological rigor, and interpretation need to be addressed before publication.
Major comments:
- The manuscript claims CCNB1 as a novel biomarker, but multiple cited studies (e.g., refs [9, 12, 63]) have already reported CCNB1’s role in CRC. Clearly articulate what is new compared to previous studies.
- The 8-gene prognostic signature shows modest predictive accuracy (AUC: 0.64–0.71), which may limit clinical utility. Discuss limitations of the model and compare with existing CRC prognostic signatures (e.g., refs [52–56]). Consider external validation using an independent dataset or cross-validation to strengthen reliability.
- The discussion states CCNB1 correlates with improved prognosis, which contradicts its known oncogenic role in other cancers. Provide mechanistic hypotheses or literature support for this paradox. Clarify whether CCNB1 acts as a stage-specific marker or interacts with other pathways in CRC.
- Functional assays (siRNA knockdown in SW480 cells) are limited to proliferation and apoptosis. No rescue experiments or downstream pathway analysis were performed. Include additional validation (e.g., CDK1 activity, cell cycle checkpoint markers). If not feasible, acknowledge this as a limitation.
- Only one CRC tumor sample was analyzed for scRNA-seq, which limits generalizability. State this limitation explicitly and consider adding publicly available scRNA-seq datasets for validation.
- Multiple testing correction for DEG identification and enrichment analysis is not mentioned. Confirm use of FDR-adjusted p-values for GO/KEGG enrichment. Report adjusted p-values in figures/tables.
Minor Comments
- Include specific AUC values for diagnostic performance and clarify CCNB1’s prognostic direction in the abstract.
- Make sure that the figure labels are readable. Figures S1–S9 are referenced but not described in detail; ensure captions are self-explanatory.
- Minor grammatical issues (e.g., “a investigation”). Improve flow in discussion by grouping findings and implications logically.
Author Response
|
Response to Reviewer 1 Comments
|
||
|
1. Summary |
|
|
|
Thank you for your patient work related to our manuscript: “Integrative bioinformatics and experimental validation establish CCNB1 as a potential biomarker for diagnosis and prognosis in colorectal cancer.” We have read the journal requirements and reviewer comments carefully. Changes corresponding to the comments are marked by red font in the revised manuscript. Below is the response to the comments and detailed description of the changes. |
||
|
2. Questions for General Evaluation |
Reviewer’s Evaluation |
Response and Revisions |
|
Does the introduction provide sufficient background and include all relevant references? |
Yes/Can be improved/Must be improved/Not applicable |
|
|
Is the research design appropriate? |
Yes/Can be improved/Must be improved/Not applicable |
In the final paragraph of the Discussion section, we elaborated on the study's limitations and directions for future research. |
|
Are the methods adequately described? |
Yes/Can be improved/Must be improved/Not applicable |
|
|
Are the results clearly presented? |
Yes/Can be improved/Must be improved/Not applicable |
We have revised several figures and their corresponding captions to more clearly elaborate the findings. |
|
Are the conclusions supported by the results? |
Yes/Can be improved/Must be improved/Not applicable |
We have revised the Discussion section by organizing it into thematic sections based on logical flow. Each theme presents the results and their corresponding conclusions, with support from the associated literature. |
|
Are all figures and tables clear and well-presented? |
Yes/Can be improved/Must be improved/Not applicable |
We have increased the font size across all figures to improve readability, with the exception of Figure 2. |
|
3. Point-by-point response to Comments and Suggestions for Authors |
||
|
Comments and Suggestions for Authors This manuscript explores the identification of CCNB1 as a diagnostic and prognostic biomarker for colorectal cancer (CRC) through integrative bioinformatics analysis and experimental validation. The study combines GEO and TCGA datasets, constructs a prognostic model, and validates CCNB1’s functional role using in vitro assays. The topic is relevant to molecular oncology and aligns with the journal’s scope, given its emphasis on molecular mechanisms and translational potential. However, several critical issues regarding novelty, methodological rigor, and interpretation need to be addressed before publication. Major comments: Comments 1: The manuscript claims CCNB1 as a novel biomarker, but multiple cited studies (e.g., refs [9, 12, 63]) have already reported CCNB1’s role in CRC. Clearly articulate what is new compared to previous studies. Response 1: Thanks for your suggestions. In the penultimate paragraph of the Discussion section, we describe the innovative aspects and significance of our work. It is revised to “In comparison with previous studies, a key advancement of our work is the comprehensive assessment of CCNB1's dual clinical utility. Our analysis demonstrated the exceptional diagnostic power of CCNB1, achieving an AUC of 0.944. More importantly, we incorporated CCNB1 into a multi-gene prognostic signature and validated its independent value for patient risk stratification using survival analysis, which provides a more nuanced tool beyond mere expression correlation. In addition, a significant contribution that distinguishes our work is the experimental validation of CCNB1's biological function in CRC progression. Through in vitro experiments, we provided direct evidence that CCNB1 promotes CRC cell proliferation, thereby bridging the gap between bioinformatic prediction and biological causality.”
Comments 2: The 8-gene prognostic signature shows modest predictive accuracy (AUC: 0.64–0.71), which may limit clinical utility. Discuss limitations of the model and compare with existing CRC prognostic signatures (e.g., refs [52–56]). Consider external validation using an independent dataset or cross-validation to strengthen reliability. Response 2: Thanks for your suggestions. We acknowledge that the AUC values ranging from 0.64 to 0.71 indicate a modest but statistically significant predictive accuracy. It is important to note that predicting long-term CRC survival with high accuracy (AUC >0.8) based solely on transcriptomics is challenging, given the profound influence of clinical and pathological determinants on patient outcomes. Many established multi-gene signatures with clinical relevance in CRC also operate within this performance range (AUC values: 0.6~0.8) while still providing valuable clinical insights [52-56]. In the final paragraph of the Discussion section, we elaborated on the study's limitations and directions for future research. It is revised to “Despite limitations including the modest predictive accuracy, the confinement of CCNB1 validation to proliferation/apoptosis, and restricted generalizability from a single scRNA-seq sample, this study provides valuable insights into CRC clinical management. In future research, we will prioritize collecting all available clinicopathological data (such as TNM stage and CEA levels), which will be integrated with genetic signatures to construct a more robust predictive model with enhanced performance. Additional validation including downstream pathway activity (e.g., CDK1 activation or key cell cycle checkpoint markers) should be incorporated to provide a more comprehensive understanding of CCNB1's role. Moreover, larger cohorts are warranted to validate and extend our observations.” After dividing the data into training and test sets, we performed time-dependent ROC curve analysis to validate the prognostic signature. For the training set, the AUC values for survival prediction were 0.60 (1-year), 0.61 (3-year), and 0.71 (5-year), respectively (Figure 4b). For the test set, the AUC values for survival prediction were 0.71 (1-year), 0.65 (3-year), and 0.72 (5-year), respectively (Figure 4c).
Comments 3: The discussion states CCNB1 correlates with improved prognosis, which contradicts its known oncogenic role in other cancers. Provide mechanistic hypotheses or literature support for this paradox. Clarify whether CCNB1 acts as a stage-specific marker or interacts with other pathways in CRC. Response 3: Thanks for your suggestions. Through literature review, we found that CCNB1 is indeed associated with poor prognosis in some cancer types, while in others (including CRC) it correlates with improved prognosis. However, the molecular mechanisms underlying the association between CCNB1 and improved prognosis in CRC remain unclear. We have added a new reference (Reference 69) on pan-cancer analysis of CCNB1, and the original Reference 69 has been renumbered as Reference 70. In the Discussion section, we included examples illustrating the relationship between CCNB1 and prognosis across different cancer types. It is revised to “Previous studies have investigated the relationships between the CCNB1 expression and prognosis in various cancers [60, 67-69]. CCNB1 overexpression correlates with adverse prognosis in diverse tumors, notably hepatocellular carcinoma, breast cancer, and bladder cancer [60, 66, 69]. Conversely, studies in lymphoma, thymoma and CRC have reported favorable outcomes associated with elevated CCNB1 expression [65-66, 69-70], highlighting its tumor-type-specific prognostic role. While an association between CCNB1 expression and improved prognosis in CRC has been established, the underlying molecular mechanisms remain incompletely understood, thus representing a key focus for future studies.”
Comments 4: Functional assays (siRNA knockdown in SW480 cells) are limited to proliferation and apoptosis. No rescue experiments or downstream pathway analysis were performed. Include additional validation (e.g., CDK1 activity, cell cycle checkpoint markers). If not feasible, acknowledge this as a limitation. Response 4: Thanks for your suggestions. We acknowledge the significant value of rescue experiments in establishing a causal relationship. However, due to current experimental condition and period constraints, we were unable to complete this validation in the current revision. In the final paragraph of the Discussion section, we elaborated on the study's limitations and directions for future research. It is revised to “Despite limitations including the modest predictive accuracy, the confinement of CCNB1 validation to proliferation/apoptosis, and restricted generalizability from a single scRNA-seq sample, this study provides valuable insights into CRC clinical management. In future research, we will prioritize collecting all available clinicopathological data (such as TNM stage and CEA levels), which will be integrated with genetic signatures to construct a more robust predictive model with enhanced performance. Additional validation including downstream pathway activity (e.g., CDK1 activation or key cell cycle checkpoint markers) should be incorporated to provide a more comprehensive understanding of CCNB1's role. Moreover, larger cohorts are warranted to validate and extend our observations.”
Comments 5: Only one CRC tumor sample was analyzed for scRNA-seq, which limits generalizability. State this limitation explicitly and consider adding publicly available scRNA-seq datasets for validation. Response 5: Thanks for your suggestions. We also attempted to utilize other publicly available single-cell datasets; however, challenges in cell clustering and annotation prevented us from achieving satisfactory results, thus limiting their utility for further validation. Therefore, we decided not to incorporate additional public datasets for validation at this stage. In the revised manuscript, we have explicitly acknowledged the limitation of the single-sample analysis and have supplemented the Discussion section with descriptions of these limitations and potential directions for future research. It is revised to “Despite limitations including the modest predictive accuracy, the confinement of CCNB1 validation to proliferation/apoptosis, and restricted generalizability from a single scRNA-seq sample, this study provides valuable insights into CRC clinical management. In future research, we will prioritize collecting all available clinicopathological data (such as TNM stage and CEA levels), which will be integrated with genetic signatures to construct a more robust predictive model with enhanced performance. Additional validation including downstream pathway activity (e.g., CDK1 activation or key cell cycle checkpoint markers) should be incorporated to provide a more comprehensive understanding of CCNB1's role. Moreover, larger cohorts are warranted to validate and extend our observations.”
Comments 6: Multiple testing correction for DEG identification and enrichment analysis is not mentioned. Confirm use of FDR-adjusted p-values for GO/KEGG enrichment. Report adjusted p-values in figures/tables. Response 6: Thanks for your suggestions. In the process of identifying DEGs, both p-values and adjusted p-values were calculated. However, this study employed a combined threshold of a p-value < 0.05 and |log2FC| > 1 to identify common GEDs present across all three datasets. Consequently, since the p-value was used in the screening process of DEGs, no specific mention was made of the adjusted p-value. Due to the limited number of significantly enriched KEGG terms identified when using the adjusted p-value as a threshold, our GO/KEGG enrichment analysis was principally conducted based on the p-value. We have included Supplementary Tables S2 and S3 to present the GO and KEGG enrichment results, which contain the outcomes of hypothesis testing and multiple testing correction, including p-values, adjusted p-values, and q-values. The previously used Tables S2 and S3 have been revised to Tables S4 and S5, respectively.
Minor Comments Comments 7: Include specific AUC values for diagnostic performance and clarify CCNB1’s prognostic direction in the abstract. Response 7: Thanks for your suggestions. We have revised the abstract. It is revised to “Colorectal cancer (CRC) is a prevalent and lethal malignancy worldwide. Despite extensive research, core genes for diagnosis and prognosis in CRC remain incompletely defined. This study aims to identify novel gene biomarkers for CRC diagnosis and prognosis based on the GEO and TCGA datasets. Integration of TCGA and GEO datasets revealed 197 common differentially expressed genes (DEGs) between CRC tumor and normal samples. Functional enrichment analysis implicated these DEGs in biological processes and signaling pathways critical to CRC progression, including cell cycle regulation and nuclear division. Protein-protein interaction (PPI) network analysis identified 17 hub genes from DEGs, including TROAP, CDKN3, CDCA3, UBE2C, CEP55, KIF11, CDC20, CCNA2, MCM4, CKS2, POLE2, MAD2L1, CCNB1, PTTG1, TPX2, TOP2A, and DLGAP5. All 17 hub genes demonstrated high diagnostic value (AUC > 0.85), including CCNB1 (AUC = 0.944). Based on the Cox proportional hazards regression, an 8-gene prognostic signature (CLCA1, CCNB1, TPM2, MMP3, AOC3, CRYAB, CA4, GUCA2A) effectively stratified patients by survival risk, with a 5-year AUC of 0.71. In vitro, CCNB1 knockdown triggered cell cycle arrest, thereby suppressing the proliferation of colorectal cancer cells. This study validated CCNB1 as a dual-purpose biomarker for CRC diagnosis and improved prognosis, highlighting its potential utility in clinical management.”
Comments 8: Make sure that the figure labels are readable. Figures S1–S9 are referenced but not described in detail; ensure captions are self-explanatory. Response 8: Thanks for your suggestions. We have revised the titles of all supplementary figures, with the exception of Figure S4, as detailed below. Figure S1: Volcano plots of DEGs between CRC and normal tissue from (a) GSE74602, (b) GSE103512, and (c) TCGA-COAD datasets. DEGs were defined by |log₂(fold change)| > 1 and p-value < 0.05. Upregulated genes are highlighted in red, downregulated genes in blue, and non-significant genes in gray. Figure S2: Bubble plots displaying the top 5 enriched GO terms for each category, derived from (a) upregulated and (b) downregulated DEGs. Figure S3: PPI network analysis (a) Visualization of the PPI network. Upregulated DEGs are shown in red and downregulated DEGs in blue. (b) PPI network showing the 17 hub genes marked in yellow. Figure S5: Kaplan-Meier survival analysis of 8 prognostic genes assessed by the log-rank test. Figure S6: Heatmap showing the expression of the top 10 marker genes across cell clusters in a CRC scRNA-seq dataset. Figure S7: Violin plot of CCNB1 expression across cell clusters in a CRC scRNA-seq dataset. Figure S8: Violin plots of S-phase and G2M-phase scores across cell clusters in a CRC scRNA-seq dataset. Figure S9: Cell cycle analysis of SW480 cells by flow cytometry in the three groups: (a) control. (b) si-NC. (C) si-CCNB1.
Comments 9: Minor grammatical issues (e.g., “a investigation”). Improve flow in discussion by grouping findings and implications logically. Response 9: Thanks for your suggestions. Sorry, there was an error here. We have corrected "a investigation" to "an investigation". We have revised the Discussion section by reorganizing it into four logically connected themes: starting from the core discovery (hub genes) to clinical translation (prognostic model), followed by experimental validation (CCNB1), and concluding with an in-depth discussion. Within each theme, we integrated relevant content previously scattered across different sections and refined the language of certain sentences to enhance clarity and coherence. The four themes are revised as follows: 4.1. Identification of 17 hub genes with high diagnostic potential in CRC pathogenesis, 4.2. An 8-gene prognostic risk model for survival prediction, 4.3. Functional validation of the prognostic hub gene CCNB1, 4.4. Research innovations, limitations, and future research directions.
|
||
|
4. Response to Comments on the Quality of English Language |
||
|
Point 1: The English could be improved to more clearly express the research. |
||
|
Response 1: After a full-text check, we optimized some grammar and language issues to enhance their comprehensibility. The revisions are presented as follows: l Line11-Line12: “Despite extensive research, core genes for diagnosis and prognosis in CRC remain incompletely defined.” revised to “Despite extensive research, core genes for diagnosis and prognosis in CRC remain to be fully elucidated.” l Line31-Line32: “In terms of mortality, it claims the position of the second foremost cause of death from cancer.” revised to “In terms of mortality, it ranks as the second leading cause of death from cancer.” l Line47-Line49: “Dysregulation of specific genes is known to critically contribute to CRC pathogenesis by altering cellular processes such as proliferation, and metastasis, ultimately diminishing patient survival.” revised to “Dysregulation of specific genes contributes critically to CRC pathogenesis by altering essential cellular processes, including proliferation and metastasis, thereby reducing patient survival.” l Line81-Line82: “This study analyzed gene expression profiles from TCGA and GEO repositories for the detection of DEGs by comparing CRC tumors with adjacent normal tissues.” revised to “This study analyzed gene expression profiles from TCGA and GEO databases to identify DEGs by comparing CRC tumors with adjacent normal tissues.” l Line90-Line92: “The GSE103512 dataset comprises 7 CRC tissues and 7 adjacent normal samples, while the GSE74602 dataset contains 30 CRC tissues and 30 adjacent normal samples.” revised to “The GSE103512 dataset comprised 7 CRC tissues and 7 adjacent normal samples, while the GSE74602 dataset contained 30 CRC tissues and 30 adjacent normal samples.” l Line108-Line110: “Enrichment analysis of DEGs was implemented based on the GO and KEGG frame-works by employing the clusterProfiler package (v4.14.6) [18], followed by visualization of results accomplished via the ggplot2 package (v3.5.1) in R v4.4.1.” revised to “We performed functional enrichment analysis of DEGs on the GO and KEGG databases with the clusterProfiler R package (v4.14.6) [18] and visualized the results using ggplot2 (v3.5.1) in R v4.4.1.” l Line117-Line119: “We evaluated the efficacy for diagnosis of these genes by conducting ROC curve analysis, determining their ability to discriminate between CRC tumor and normal samples.” revised to “We evaluated the diagnostic efficacy of these genes by conducting ROC curve analysis, determining their ability to discriminate between CRC tumor and normal samples.” l Line259-Line262: “Time-dependent ROC curves validated the prognostic signature, yielding AUC values of 0.64 (1-year), 0.63 (3-year), and 0.71 (5-year) for survival prediction (Figure 4b).” revised to “Time-dependent ROC curves validated the prognostic signature, with AUC values of 0.60 (1-year), 0.61 (3-year), and 0.71 (5-year) in the training set (Figure 4b), and 0.71 (1-year), 0.65 (3-year), and 0.72 (5-year) in the test set (Figure 4c).” l Line310-Line311: “Our study integrated two GEO microarray datasets and TCGA RNA-seq data, identifying 197 DEGs (68 upregulated, 129 downregulated) between CRC and normal tissues.” revised to “Our study identified 197 DEGs (68 upregulated, 129 downregulated) between CRC and normal tissues by integrating two GEO microarray datasets and TCGA RNA-seq data.” l Line312-Line315: “Functional enrichment analysis revealed upregulated DEGs involvement in nuclear division, nuclear chromosome segregation, sister chromatid segregation, microtubule cytoskeleton organization involved in mitosis, spindle, and spindle pole.” revised to “Functional enrichment analysis revealed that the upregulated DEGs were significantly enriched in nuclear division, nuclear chromosomal segregation, sister chromatid segregation, and microtubule cytoskeleton organization involving spindle and spindle pole formation.” l Line318-Line320: “These findings demonstrate that these DEGs function through multiple signaling path-ways and contribute to CRC oncogenesis.” revised to “These findings collectively demonstrated that the upregulated DEGs drive CRC oncogenesis through the activation of multiple proliferative signaling pathways.” l Line369-Line370: “Significant divergence in clinical outcomes was observed between the high- and low-risk cohorts.” revised to “This model effectively stratified patients into distinct high- and low-risk cohorts with significant divergence in clinical outcomes.”
|
||

Reviewer 2 Report
Comments and Suggestions for Authors
The aim of this study was to identify novel gene biomarkers for colorectal cancer diagnosis and prognosis on the basis of gene expression omnibus and the cancer genome atlas datasets. Integration of the datasets revealed 197 differently expressing genes between colorectal cancer and normal samples. Protein-protein interaction network analysis identified 17 hub genes including the CCNB1 gene. In vitro, knockdown of CCNB1 triggered cell cycle arrest. thereby suppressing the proliferation of colorectal cancer cells. The study validated CCNB1 as a dual-purpose biomarker for colorectal cancer diagnosis and prognosis, highlighting its potential utility in clinical management of colorectal cancer.
The article can be accepted for publication after a minor revision. Two points should be addressed:
1) The Authors concluded that the overexpression of the CCNB1 was correlated with improved prognosis in colorectal cancer patients. This is surprsising, since an elevated expression of CCNB1 observed in diverse cancers promoted uncontrolled cancer cell proliferation. Would it mean that an overexpression of the CCNB1 may be good or bad for patients depending on the type of cancer ?
2) Minor point: letters in Figures 1-6 should be enlarged, for clarity,
Author Response
|
Response to Reviewer 2 Comments
|
||
|
1. Summary |
|
|
|
Thank you for your patient work related to our manuscript: “Integrative bioinformatics and experimental validation establish CCNB1 as a potential biomarker for diagnosis and prognosis in colorectal cancer.” We have read the journal requirements and reviewer comments carefully. Changes corresponding to the comments are marked by red font in the revised manuscript. Below is the response to the comments and detailed description of the changes. |
||
|
2. Questions for General Evaluation |
Reviewer’s Evaluation |
Response and Revisions |
|
Does the introduction provide sufficient background and include all relevant references? |
Yes/Can be improved/Must be improved/Not applicable |
|
|
Is the research design appropriate? |
Yes/Can be improved/Must be improved/Not applicable |
|
|
Are the methods adequately described? |
Yes/Can be improved/Must be improved/Not applicable |
|
|
Are the results clearly presented? |
Yes/Can be improved/Must be improved/Not applicable |
We have revised several figures and their corresponding captions to more clearly elaborate the findings. |
|
Are the conclusions supported by the results? |
Yes/Can be improved/Must be improved/Not applicable |
We have revised the Discussion section by organizing it into thematic sections based on logical flow. Each theme presents the results and their corresponding conclusions, with support from the associated literature. |
|
Are all figures and tables clear and well-presented? |
Yes/Can be improved/Must be improved/Not applicable |
We have increased the font size across all figures to improve readability, with the exception of Figure 2. |
|
3. Point-by-point response to Comments and Suggestions for Authors |
||
|
Comments and Suggestions for Authors The aim of this study was to identify novel gene biomarkers for colorectal cancer diagnosis and prognosis on the basis of gene expression omnibus and the cancer genome atlas datasets. Integration of the datasets revealed 197 differently expressing genes between colorectal cancer and normal samples. Protein-protein interaction network analysis identified 17 hub genes including the CCNB1 gene. In vitro, knockdown of CCNB1 triggered cell cycle arrest. thereby suppressing the proliferation of colorectal cancer cells. The study validated CCNB1 as a dual-purpose biomarker for colorectal cancer diagnosis and prognosis, highlighting its potential utility in clinical management of colorectal cancer. The article can be accepted for publication after a minor revision. Two points should be addressed: Comments 1: The Authors concluded that the overexpression of the CCNB1 was correlated with improved prognosis in colorectal cancer patients. This is surprising, since an elevated expression of CCNB1 observed in diverse cancers promoted uncontrolled cancer cell proliferation. Would it mean that an overexpression of the CCNB1 may be good or bad for patients depending on the type of cancer? Response 1: Thanks for your suggestions. Through literature review, we found that CCNB1 is indeed associated with poor prognosis in some cancer types, while in others (including CRC) it correlates with improved prognosis. However, the molecular mechanisms underlying the association between CCNB1 and improved prognosis in CRC remain unclear. In the Discussion section, we included examples illustrating the relationship between CCNB1 and prognosis across different cancer types. It is revised to “Previous studies have investigated the relationships between the CCNB1 expression and prognosis in various cancers [60, 67-69]. CCNB1 overexpression correlates with adverse prognosis in diverse tumors, notably hepatocellular carcinoma, breast cancer, and bladder cancer [60, 66, 69]. Conversely, studies in lymphoma, thymoma and CRC have reported favorable outcomes associated with elevated CCNB1 expression [65-66, 69-70], highlighting its tumor-type-specific prognostic role. While an association between CCNB1 expression and improved prognosis in CRC has been established, the underlying molecular mechanisms remain incompletely understood, thus representing a key focus for future studies.” Comments 2: Minor point: letters in Figures 1-6 should be enlarged, for clarity, Response 2: Thanks for your suggestions. We have increased the font size across all figures to improve readability, with the exception of Figure 2. Specifically, the font in Figure 2a was not enlarged to avoid overlap of the gene labels.
|
||
|
4. Response to Comments on the Quality of English Language |
||
|
Point 1: none |
||
|
Response 1: none |
||